# Analysis of Light Intensity and Charge Holding Time Dependence of Pinned Photodiode Full Well Capacity

**DOI:** 10.3390/s23218847

**Published:** 2023-10-31

**Authors:** Ken Miyauchi, Toshiyuki Isozaki, Rimon Ikeno, Junichi Nakamura

**Affiliations:** 1Brillnics Japan Inc., 6-21-12, Minami-Oi, Shinagawa-ku, Tokyo 140-0013, Japan; isozaki.toshiyuki@brillnics.com (T.I.); ikeno.rimon@brillnics.com (R.I.); nakamura.junichi@brillnics.com (J.N.); 2Graduate School of Engineering, Tohoku University, 6-6-05, Aza-Aoba, Aramaki, Aoba-ku, Sendai 980-8579, Japan

**Keywords:** CMOS image sensor, buried overflow path, full well capacity, light intensity, charge holding time, equilibrium PDFWC

## Abstract

In this paper, the light intensity and charge holding time dependence of pinned photodiode (PD) full well capacity (FWC) are studied for our pixel structure with a buried overflow path under the transfer gate. The formulae for PDFWC derived from a simple analytical model show that the relation between light intensity and PDFWC is logarithmic because PDFWC is determined by the balance between the photo-generated current and overflow current under the bright condition. Furthermore, with using pulsed light before a charge holding operation in PD, the accumulated charges in PD decrease with the holding time due to the overflow current, and finally, it reaches equilibrium PDFWC. The analytical model has been successfully validated by the technology computer-aided design (TCAD) device simulation and actual device measurement.

## 1. Introduction

Complementary metal-oxide-semiconductor (CMOS) image sensors have been used in many applications, and photodiode full well capacity (PDFWC) is one of the important parameters because the maximum handling charge is determined by it in general.

Recently, to achieve over 100 dB dynamic range (DR), a lateral overflow integration capacitor (LOFIC) scheme [1,2,3,4,5,6] and a triple quantization digital pixel sensor [7,8,9] have been developed. To achieve high DR (HDR), a couple of signal readout modes with different pixel gains are implemented in these sensors, and one of the factors to define the performance at the mode transition points, such as the signal-to-noise ratio and linearity, is PDFWC [3]. Therefore, PDFWC is an important parameter not only for the maximum handling charge but also for the mode transition performance.

In almost all the recent CMOS image sensors, P+NP pinned PD is used as the photo detector [10,11,12]. It has been observed that PDFWC is dependent on the light intensity and transfer gate potential [13]. Although, in our HDR sensor development [7,8,9], we have found that PDFWC also changes with light intensity, as shown in Figure 1a, as it has different behavior with respect to the transfer gate voltage during integration compared to [13]. Also, we have shown that PDFWC decreases as the time interval between the end of pulsed light exposure and the start of signal readout [14], which is referred to as the charge holding time, increases, as shown in Figure 1b. Equivalent pixel circuit [7,8,9] schematic and pulse timing diagrams to obtain the results, shown in Figure 1a,b, are shown in Figure 2. The differences between the conventional rolling shutter pixel and this pixel are the buried overflow path under the TG and the in-pixel current source. Although the current source is equipped to perform the global shutter operation, we have confirmed that it does not affect our PD analysis.

Some analytical models for such light intensity dependence of PDFWC have been proposed in the literature [13,14,15,16,17]. In [18,19], a measurement method of the PD potential with charge injection is proposed, and the amount of the overflow charge is measured directly. Then, the intrinsic PDFWC is measured with the same method in [20]. In [18] and [21], the potential barrier height of the PD well analysis is also mentioned.

Although similar models in [13,14,15,16,17] are used in this work, minor modifications are needed for our pixel [7,8,9] because of its unique structure with the buried overflow path under the transfer gate [4,5,6]. Figure 3 shows the PD cross-section and potential of the buried overflow path, which is formed by the low concentration n-layer under TG. Generally, to prevent the dark current at the Si surface under TG, a negative TG bias voltage is applied [22,23,24], and the surface potential becomes low. On the other hand, electrons above the PD saturation level overflow to FD through the buried overflow path because the potential of it is higher than that of the surface region. As a result, both good overflow efficiency and low dark current can be achieved in this structure. In this work, the potential barrier height of the buried overflow path is not affected by the TG-negative bias voltage because it is formed deep enough in the Si bulk region. It is different from the models in [13,14,15,16,17] because TG voltage is not a parameter anymore. Also, our TCAD simulation shows the parameters of equations and the actual potential distribution. We confirmed that it shows good evidence of the analytical model, which was not described in [13,14,15,16,17].

The remainder of this paper is organized as follows: The light intensity and charge holding time dependence phenomena of PDFWC were introduced in Section 1. In Section 2, our analytical model and formulization of these phenomena are described. Comparison results of these analytical models, TCAD simulation, and measurement are shown in Section 3, and conclusions are provided in Section 4.

## 2. Analytical Model

In this section, our analytical model [25] and formulation of light intensity and charge holding time dependence of PDFWC are described.

### 2.1. Light Intensity Dependence

Figure 4 shows potential and charge distribution changes from PD reset to PD saturation based on [26]. The fully depleted condition after the electronic shutter is shown in Figure 4a, where no electrons exist inside the PD, and the relationship $E_{f}=E_{i}$ holds. Figure 4b shows a low illuminance saturation case. Here, the PD saturation is determined by the balanced condition between the input and output currents of the PD. Therefore, the intrinsic PD current, which flows from the p-layer to the n-layer, balances with the overflow current (evaporation or thermionic current) in Figure 4b because of the low light intensity, and thus there is a small photo-current. When the light intensity is high, as shown in Figure 4c, i.e., in a high light illuminance saturation case, the photo-generated current balances with the overflow current.

The simple PD model for theoretical analysis and the parameters are shown in Figure 5, where three components, the photo-generated current $I_{ph}$, the intrinsic PD current $I_{PPD}$, and the overflow current $I_{of}$, are considered. According to some previous studies [14,18], the maximum fully depleted PD potential $V_{pin}$ has temperature dependence, and the overflow potential $V_{of}$ strongly depends on the FD voltage. On the other hand, in our analysis, $V_{pin}$, $V_{of}$, and the FD voltage are assumed constant to simplify the model.

First, these three current components are formulated. The photo-generated current $I_{ph}$ is formulated as
(1)$$I_{ph}=q\cdot R\cdot P$$
where q, R, and P denote the unit electron charge, the responsivity in [e^−^/lx·s], and the face-plate illuminance in [lx], respectively.

Next, the intrinsic PD current $I_{PPD}$ is formulated by the general PN junction diode current equation as
(2)$$I_{PPD}=I_{sat}\cdot \left\{e^{\left(\frac{-V_{S}}{V_{T}}\right)}-1\right\}$$
where $I_{sat}$, $V_{S}$, and $V_{T}$ are the PN junction reverse bias saturation current, the PD potential, and the thermal voltage ($=\frac{kT}{q}$), respectively. If a strong reverse bias condition, $\left|V_{S}\right|\gg \left|V_{T}\right|$, is assumed in Equation (2), $I_{PPD}$ can be approximated by $-I_{sat}$.

Finally, the overflow current $I_{of}$ is formulated as
(3)$$I_{of}=I_{0}\cdot e^{\left(\frac{-∆V_{b}}{n\cdot V_{T}}\right)}$$
where $∆V_{b}$, $I_{0}$, and n denote the barrier height between the PD and the overflow path (i.e., $V_{S}-V_{of}$), the overflow current that would flow at $∆V_{b}=0$, and the non-ideality factor, respectively.

Under the PD saturation conditions, the input and the output currents should be balanced. Therefore, the following relationship holds.
(4)$$I_{ph}=I_{of}+I_{PPD}$$

From Equations (1) to (4), the minimum PD potential under PD saturation is given by
(5)$$V_{S\_min}=V_{of}-n\cdot V_{T}\cdot ln\left(\frac{I_{sat}}{I_{0}}+\frac{qR}{I_{0}}P\right)$$

Equation (5) shows that $V_{S\_min}$ decreases logarithmically to flow larger $I_{of}$ when $I_{ph}$ increases. On the other hand, PDFWC is expressed with the basic equation as
(6)$$N_{FWC}=\frac{C_{PD}}{q}\left(V_{pin}-V_{S\_min}\right)$$
where $C_{PD}$ and $V_{pin}$ are the PD capacitance and the maximum PD fully depleted potential. Here, the PD capacitance $C_{PD}$ is assumed to be constant to simplify the model.

From Equations (5) to (6), PDFWC is given by
(7)$$N_{FWC}=\frac{C_{PD}}{q}\left\{V_{pin}-V_{of}+n\cdot V_{T}\cdot ln\left(\frac{I_{sat}}{I_{0}}+\frac{qR}{I_{0}}P\right)\right\}$$

Therefore, PDFWC is a function of face-plate illuminance P, and it increases logarithmically with the face-plate illuminance.

Next, the very low illuminance saturation case shown in Figure 4b is considered. In this case, the output current is only $I_{sat}$ and the input current is $I_{of}$ because $I_{ph}\sim 0$. Therefore, $I_{sat}$ and $I_{of}$ are balancing under the saturation condition. PDFWC with this condition is defined as the equilibrium PDFWC, $N_{FWC\_eq}$. In our pixel structure, it is formulated as Equation (8) by assuming P=0 in Equation (7).
(8)$$N_{FWC\_eq}=\frac{C_{PD}}{q}\left\{V_{pin}-V_{of}+n\cdot V_{T}\cdot ln\left(\frac{I_{sat}}{I_{0}}\right)\right\}$$

In our pixel structure, the equilibrium PDFWC means PDFWC with the equilibrium condition between the n-layer potential ($V_{S}$) and the overflow barrier potential ($V_{of}$). Under this condition, $I_{sat}$ balances with $I_{of}$. These dynamic behaviors are summarized as a simple model in Figure 6.

### 2.2. Charge Holding Time Dependence

Figure 7 shows potential and charge distribution changes from PD saturation to equilibrium PDFWC with a pulsed light. After the end of pulsed light illumination, the charge holding operation in PD follows, as shown in Figure 7b. During the charge holding operation, the PD is not illuminated ($I_{ph}=0$), and the PD charges are being held in the PD.

The PD potential change during the charge holding operation is formulated as below under the condition of $I_{ph}=0$.
(9)$$\frac{dV_{S}}{dt_{hold}}=\frac{I_{out}}{C_{PD}}-\frac{I_{in}}{C_{PD}}=\frac{\left(I_{of}+I_{PPD}\right)}{C_{PD}}-\frac{I_{ph}}{C_{PD}}=\frac{I_{0}\cdot e^{\left(\frac{-∆V_{b}}{n\cdot V_{T}}\right)}-I_{sat}}{C_{PD}}$$

Solving Equation (9) yields the PD potential as
(10)$$V_{S}=V_{of}+n\cdot V_{T}\cdot \ln\left\{e^{\left(-\frac{I_{sat}}{C_{PD}\cdot n\cdot V_{T}}t_{hold}+A\right)}+\frac{I_{0}}{I_{sat}}\right\}$$
where A is a constant. With Equation (6), PDFWC is obtained as
(11)$$N_{PD}=\frac{C_{PD}}{q}\left[V_{pin}-V_{of}-n\cdot V_{T}\cdot \ln\left\{e^{\left(-\frac{I_{sat}}{C_{PD}\cdot n\cdot V_{T}}t_{hold}+A\right)}+\frac{I_{0}}{I_{sat}}\right\}\right]$$

Therefore, PDFWC is a function of the charge holding time, $t_{hold}$, which demonstrates PDFWC has charge holding time dependence.

When the condition of very long $t_{hold}$, as shown in Figure 7c, is assumed, Equation (10) becomes
(12)$$V_{S}=V_{S\_long}=V_{of}-n\cdot V_{T}\cdot \ln\left(\frac{I_{sat}}{I_{0}}\right)$$

The potential changes during this process are summarized in Figure 8. 

### 2.3. Equilibrium PDFWC

In this section, the equilibrium PDFWC is discussed. From Equations (6) to (12), PDFWC after a very long charge holding time is formulated as
(13)$$N_{PD\_long}=\frac{C_{PD}}{q}\left\{V_{pin}-V_{of}+n\cdot V_{T}\cdot \ln\left(\frac{I_{sat}}{I_{0}}\right)\right\}$$

Equation (13) is identical to Equation (8), which demonstrates that PDFWC after a very long charge holding time ($N_{PD\_long}$) also reaches $N_{FWC\_eq}$. In these conditions, $I_{of}$ balances with $I_{sat}$ under the condition of $I_{ph}\sim 0$. Therefore, the barrier height of $∆V_{b}$ is corresponding to the PN junction built-in potential between the n-layer of PD and the surrounded p-layer in the conventional PD structure. On the other hand, in our PD structure, it is defined as the barrier height between the potential of the PD and the overflow path in the equilibrium condition. The equilibrium PDFWC can be used to define the PDFWC, which is determined by only the PD structure, and it is not affected by the light intensity condition. However, it does not match the PDFWC measured in the actual case because a PD in CMOS image sensors saturates under illuminated conditions.

## 3. Experimental Validation

In this section, the experimental validation results are described. The analytical model is compared to the TCAD simulation and actual device characterization results.

### 3.1. Device Information

A digital-pixel sensor by 45 nm/65 nm stacked process [7,8,9] is used in this validation study. A chip micrograph and the specifications are shown in Figure 9.

### 3.2. Basic Characteristics of Transfer Gate Transistor

Before starting the analysis under illuminated conditions, the basic transistor characteristic was estimated using TCAD simulation, which consists of the 3D process and device simulations with actual layout.

First, the $∆V_{b}$ dependence of $I_{of}$ was analyzed. The simulation set-up, simulated PD/FD current—$∆V_{b}$ curves, and potential along the section x–x’ as a function of the PD potential $V_{S}$, are shown in Figure 10a–c, respectively. In this simulation, the constant current flows from FD to PD depending on the PD voltage, $V_{S}$. As shown in Figure 10b, $I_{of}$ flowing from FD to PD is dependent on $V_{S}$ exponentially, when $V_{S}$ is low enough and then $∆V_{b}$ is small enough. It matches the $I_{of}$ dependence on $∆V_{b}$ modeled in Equation (3). From the slope of Figure 10a, the non-ideality factor n of Equation (3) is calculated as 1.

### 3.3. Simulation and Measurement Set-Up for Light Intensity and Charge Holding Time Dependence of PDFWC

Next, the light intensity dependence and the charge holding time dependence of PDFWC were analyzed. Figure 11a shows the simulation set-up for it. In contrast to Figure 10a, the PD electrons are generated by an external light source in this simulation. Figure 11b shows the photo-response curve, the FD current, and the photo-generated current under the static light condition. In the PD saturation region, it is obvious that Equation (4) holds with $I_{of}\gg I_{PPD}$.

### 3.4. Light Intensity Dependence Validation

Figure 12a shows the light intensity dependence of the potential distribution. It clearly shows that there is light intensity dependence on the PD saturation potential.

Next, the relation between the light intensity and the PD potential $V_{S}$ in Figure 12a is shown in Figure 12b. Furthermore, the relation between the light intensity and PDFWC is shown in Figure 12c. The TCAD simulation result of Figure 12b and Equation (5) agree well, and the PD potential decreases logarithmically. The characterization result matches the simulation and the analytical model of Figure 12c and Equation (7). In Figure 12b,c, the flat level is determined by the ratio of $I_{sat}$ to $I_{0}$, as indicated by Equation (8).

### 3.5. Charge Holding Time Dependence Validation

Figure 13 compares the analytical model and TCAD simulation of the charge holding time dependence of PD potential and PD electrons. Figure 13a shows the potential distribution change from PD saturation to the equilibrium PDFWC under dark conditions after pulsed light illumination. During the pulsed light illumination, PD is oversaturated in this simulation and measurement. It is seen that the PD potential increases as the charge holding time increases, and finally, the potential reaches the equilibrium condition.

This PD potential change in Figure 13a is plotted in Figure 13b. The TCAD simulation result of (b) agrees well with Equation (10). The characterization result matches the analytical model given by Equation (11), as shown in Figure 13c.

### 3.6. Equilibrium PDFWC Validation

The potential distributions shown with the dashed gray lines in Figure 12a and Figure 13a, i.e., the case of low illuminance saturation only with dark current and the case of a long holding time of 100 s, are re-plotted in Figure 14. It is confirmed that the equilibrium PDFWC condition is reached in both cases, which reproduces Equations (8) and (13).

## 4. Conclusions

In this paper, the light intensity dependence of PDFWC and the PD charge reduction phenomenon during the charge holding operation were analyzed and formulated. They were verified with TCAD simulations and actual device characterization of our 3Q-DPS pixel with a buried overflow path. In this structure, the dependence of the low level of TG pulse can be neglected because the overflow voltage $V_{of}$ does not depend on it when it is set negative enough for surface pinning.

First, PD saturation under a static light condition was considered. During the PD saturation condition, the electrons flowing in and those flowing out from the PD should balance. Therefore, the potential barrier between the PD and the overflow path is a function of the photo-generated current. This means that it is also a function of input light intensity. It decreases logarithmically to the light intensity increase, increasing PDFWC logarithmically. 

Next, PDFWC reduction with a pulsed light was considered. During the charge holding operation after pulsed light illumination, the overflow current draws the charges from PD to FD. As a result, the PD charges reduce. Finally, the overflow current and the reverse-bias saturation current balances, and the PDFWC reduction stops.

The analytical model indicates that PDFWC converges to the equilibrium PDFWC for both cases where the light intensity is close to zero with a static light, and where the signal charges remain in the PD at dark after illuminated by a pulsed light before they are read out. Both the characterization result and the TCAD simulation result support the analytical model with the new structure of TG with a buried overflow path. In the actual case, PDFWC should be defined with consideration of these phenomena because a PD in CMOS image sensors saturates under the bright condition. Also, this modeling is important, especially for the design of HDR multiple gain mode sensors.

## Figures and Tables

**Figure 1 sensors-23-08847-f001:**
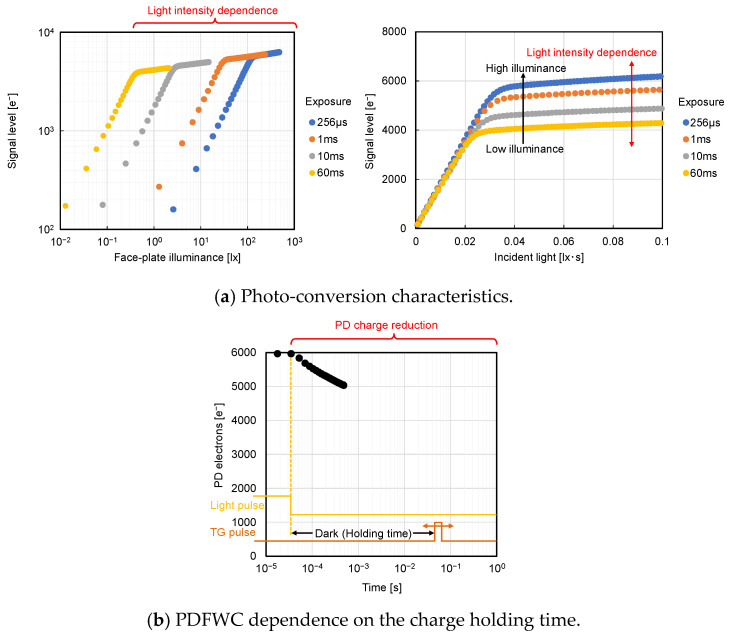
Measured dynamic behaviors of PPD.

**Figure 2 sensors-23-08847-f002:**
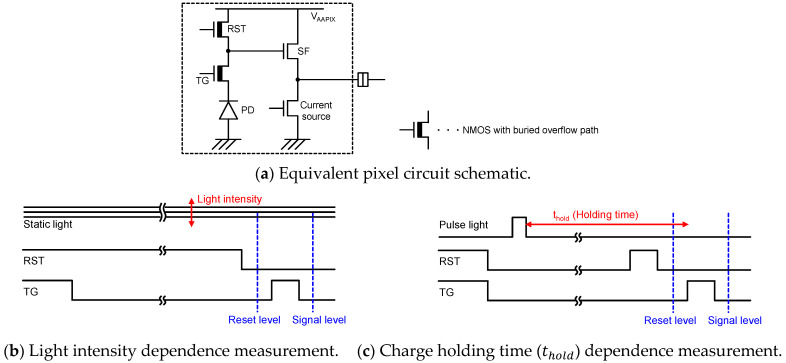
Pixel schematic and pulse timing for characterization.

**Figure 3 sensors-23-08847-f003:**
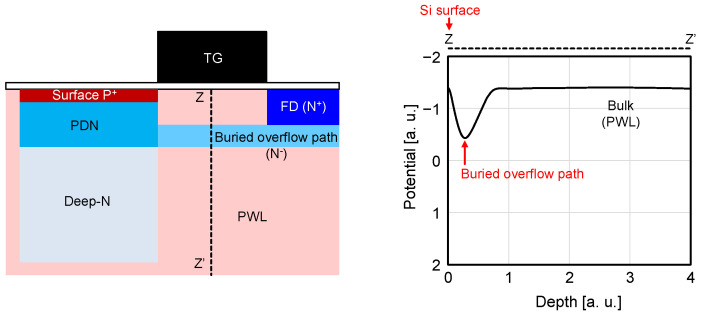
PD cross-section and potential of the buried overflow path.

**Figure 4 sensors-23-08847-f004:**
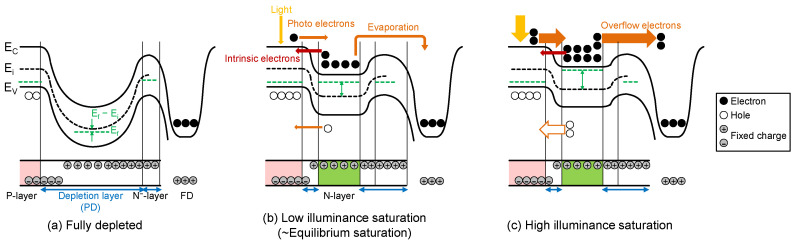
Potential and charge distribution changes from PD reset to PD saturation.

**Figure 5 sensors-23-08847-f005:**
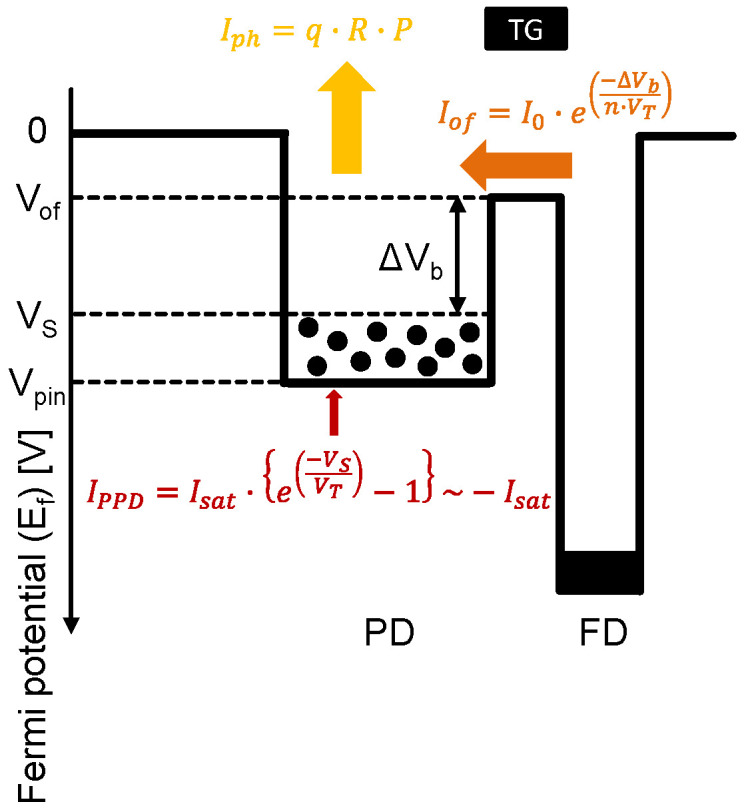
Simple PD model for theoretical analysis.

**Figure 6 sensors-23-08847-f006:**
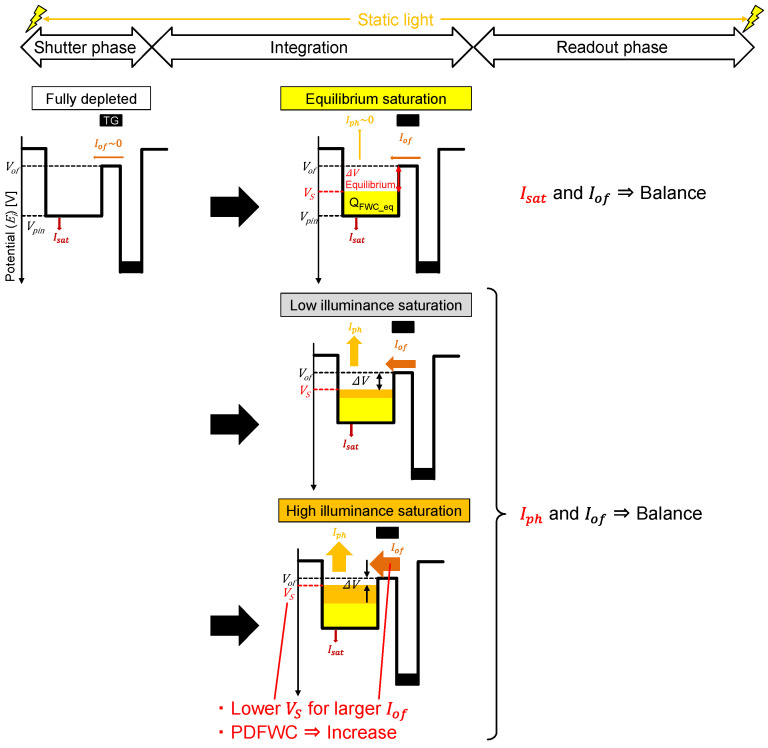
A simple model of dynamic behavior from PD reset to PD saturation under static light condition.

**Figure 7 sensors-23-08847-f007:**
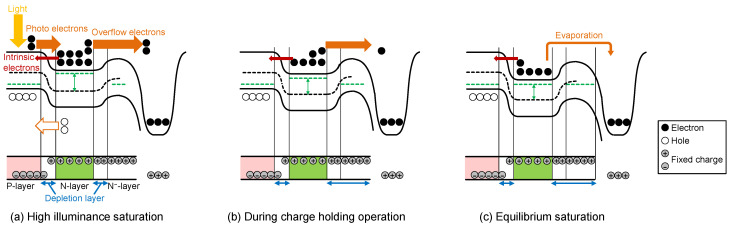
Potential and charge distribution changes from PD saturation to equilibrium PDFWC.

**Figure 8 sensors-23-08847-f008:**
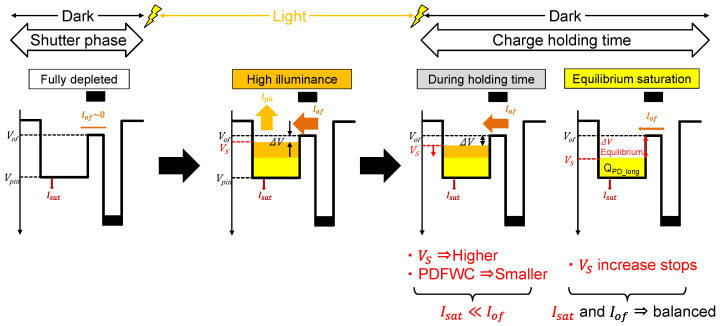
A simple model of PD charge reduction during charge holding operation with pulse light.

**Figure 9 sensors-23-08847-f009:**
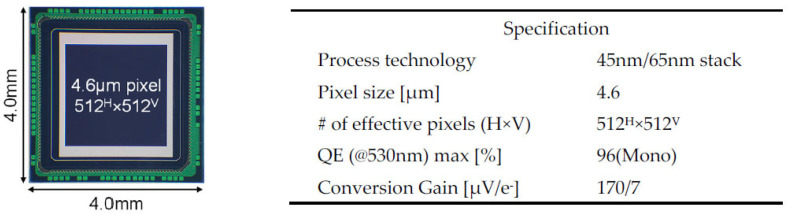
Chip micrograph and specifications of our developed stacked 3Q-DPS [7,8,9].

**Figure 10 sensors-23-08847-f010:**
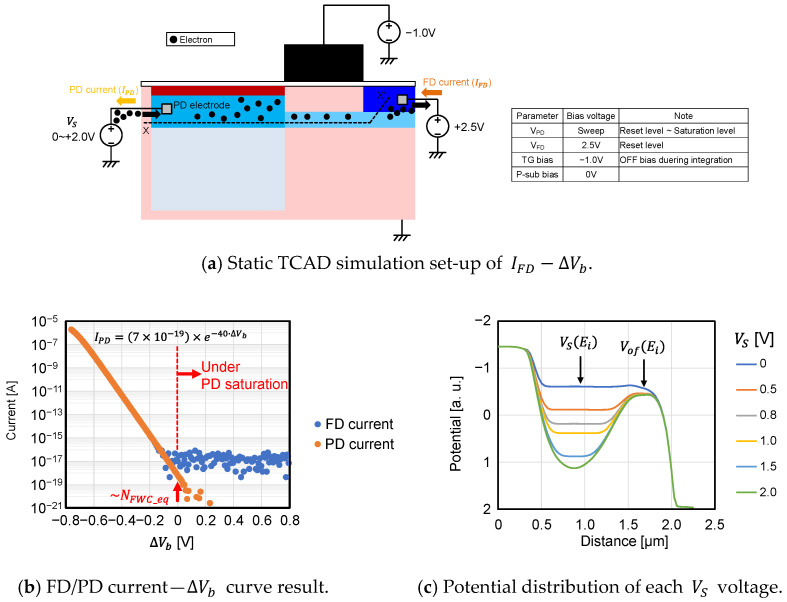
Relation between ${∆V}_{b}$ and $I_{of}$ with static TCAD simulation.

**Figure 11 sensors-23-08847-f011:**
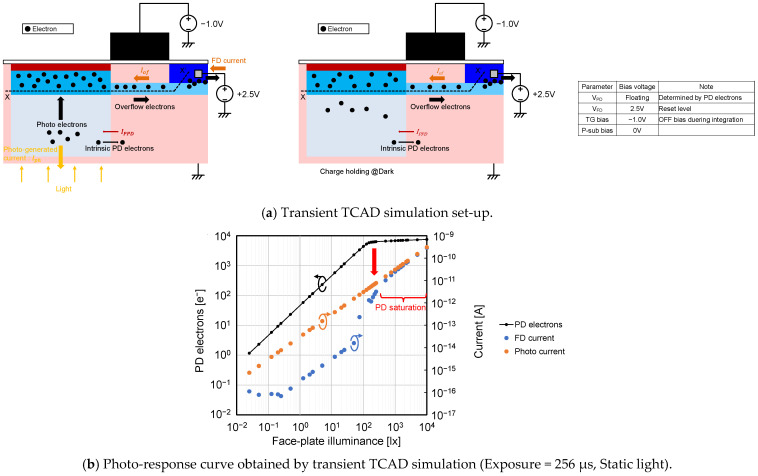
Transient TCAD simulation for light intensity and charge holding time dependence.

**Figure 12 sensors-23-08847-f012:**
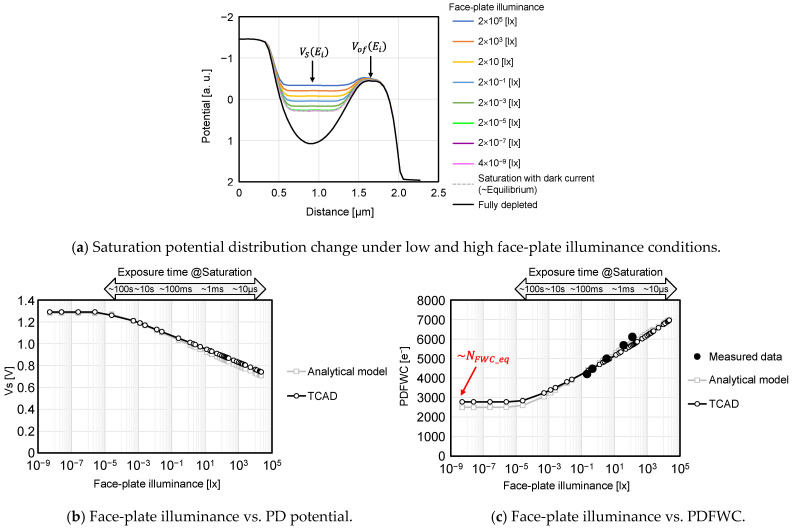
PDFWC under various light intensity conditions.

**Figure 13 sensors-23-08847-f013:**
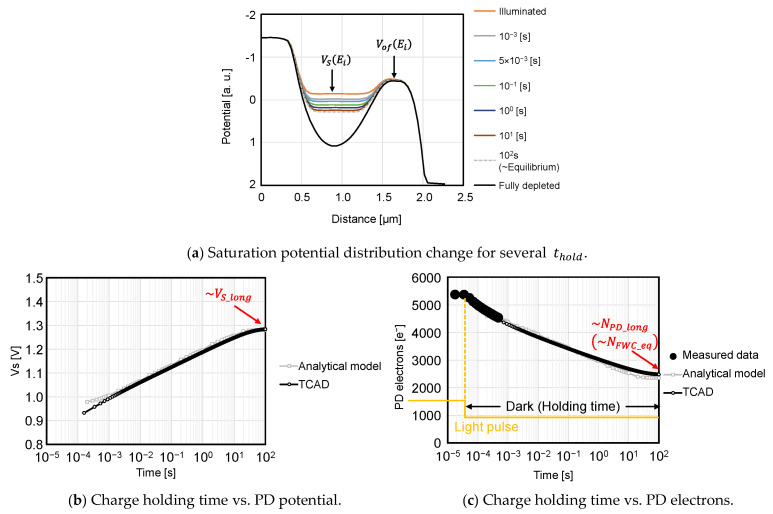
PDFWC with long charge holding times.

**Figure 14 sensors-23-08847-f014:**
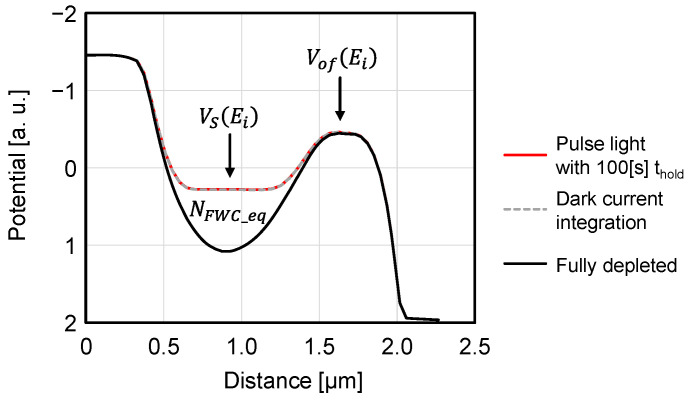
TCAD simulation results of equilibrium PDFWC potential.

## Data Availability

Not applicable.

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
