# Peer review of "Analysis of Light Intensity and Charge Holding Time Dependence of Pinned Photodiode Full Well Capacity"

_sensors, 2023, doi:10.3390/s23218847_

Round 1

Reviewer 1 Report

Comments and Suggestions for Authors

This manuscript contains rich experimental comparisons to provide suggestions for photodiode design optimization.

Comments on the Quality of English Language

The English writing in this manuscript is good and the overall description is clear and concise.

Author Response

Dear reviewer 1,

Thanks for your comments.
I modified some sentences and uploaded the new version of the manuscript.

Best regards,
Ken Miyauchi

Reviewer 2 Report

Comments and Suggestions for Authors

Author Response

Dear Reviewer 2,

Thanks for your kind feedback.
I modified the manuscript and uploaded it.

The below is detail of my reply.

-P2, line 48: the statement “Therefore, in general, PDFWC has been considered a constant value in measurement conditions.” is wrong. Please refer to ref16.
⇒I understood and agree with your comments. On the other hand, in the general case, the phenomena in ref16 are not considered in many cases. So, in the introduction phase, I would like to show this general model, then I would like to show this general model is not sufficient in some cases such as HDR sensors.

-- p3, line 95: what is new compared to other papers? A new architecture is proposed here but is not really exploited; then modeling of FWC had already been proposed in papers such as in ref16 - so whats is really new here?
⇒Thanks for your kind feedback. Our progresses are 1. New model without parameter of TG voltage for our new structure 2. It was reproduced by TCAD simulation (Actual potential distribution, current, and accumulated electrons). The TCAD demonstration, especially for the potential distribution was not described in the ref16.

-- p5, line 138: how is obtained equation (2)? Could you give a reference?
⇒Thanks for your kind comment.
I added an explanation of “Next, the intrinsic PD current I_PPD is formulated by the general PN junction diode current equation”

-- p6, line 146: how is defined the non ideality factor? What is the value here, is it a calibration parameter?
⇒Thanks for your kind feedback. It is a calibration parameter and calculated by Fig. 10. I added a sentence of “From the slope of Figure 10 (a), the non-ideality factor, n of eq. (3) is calculated as 1.” in p.9. Also, I added the actual slope value in Fig. 10 (a).

-- p7 Fig 6 does not bring any additional information
⇒Thanks for your comment. I understood your point. On the other hand, the purpose of this figure is to clarify the model which is described in the sentences. It can be helpful for readers who do not know this field to understand the concept of the model easier. So, I would like to put this figure in the paper.

-- p9, line 221: the TCAD setup is not described at all. Which tool is used? what are the initial conditions (VTG, Vdd, Vground)? In order to reproduce the pixel behavior, FD must be floating: how is it achieved in this simulation?
⇒Thanks for your kind comments. From your comment, I realized floating FD was my typo. I modified Fig. 2. The other voltage condition table was added in Fig. 11 (a). The TCAD tool is commercial 3D TCAD tool in CMOS image sensor companies, and the simulation type is static in Fig. 10, transient in Figs. 11-14. We used actual device gds and profile for the input of this simulation.

-- p9, line 223-227: how is extracted Vb and how electrons are injected into the PPD? can the authors provide a timing diagram?
⇒Thanks for your questions. This simulation is a static simulation, not transient. So, we did not use any pulse timings in this simulation. So, I modified the sentence as “The constant current flows from FD to PD depending on the PD voltage.”

-- p10, title 3,3: this title is inappropriate as this part does not show simulations with time dependence.
⇒ Thanks for your kind comment. Based on your comment, I added a figure about the simulation set-up for charge holding time dependence analysis. In this figure, I would like to show the simulation condition of floating PD with light for these analysis.

-- p10, line 239: what are the light characteristics? how long is the illumination? what are the initial conditions?
⇒ Thanks for your kind comments. The initial condition table was added in Fig. 11 (a). Also, I added the exposure condition in the figure caption.

-- p11, Fig12: what is called evaluation?
⇒Thanks for your question, and sorry for the confusion. I called “Measured actual Si data” as “evaluation” in this paper.
Then, I changed from “Evaluation” to “Measured data”

-- p11, line 270: it is not clear what the authors call a “generated current”: is it the photon current? Or something else?
⇒Thanks for your question. I removed this paragraph based on your review.

-- p11, line 271-272: this sentence and Fig 13 does not bring anything new.
⇒Thanks for your advice. I removed this paragraph based on your review to make this paper shorter.

-- p12, line 281: What are the initial conditions? Is the PD over saturated? is there a calibration of the model? How are adjusted A and n?
⇒Thanks for your comments. I added an explanation of “During the pulsed light illumination, PD is over saturated in this simulation and measurement.”. “n” is adjusted by fig. 10 (a) as shown in the former answer to your review. “A” is fitting parameter.

-- conclusion, line 323-325: this conclusion is not new and has already been demonstrated in other ref related to FWC studies
⇒Thanks for your feedback. I understood your comment, and our final conclusion is surely similar to the previous work. So, I added some explanations in conclusion, especially for the new TG structure and TCAD validation.

Round 2

Reviewer 2 Report

Comments and Suggestions for Authors

Most of my remarks have been taken into account. However I still have the feeling that this paper is too long, while it brings few novelties to the community. Actually, this work is just an update of ref 16.

In particular, the introduction should be shortened.

TCAD simulations are still not well described: it would be nice to see a real doping distribution of the device, the model activated... Authors must also indicate in the manuscript if simulations are transient or static: this is not clear at all.

Author Response

Dear reviewer2

Thanks for your additional feedback.
I updated my manuscript based on your feedback.

1. Redundant explanations were removed in all chapters.
Also, I made introduction shorter.
As a result, the number of pages reduced from 15 to 13.

2. I added simulation type for figure captions based on your comment.
But I decided not to add the real doping distribution because it is not necessary for this analysis, and I would like to keep it as confidential information.